# The Mutational Robustness of the Genetic Code and Codon Usage in Environmental Context: A Non-Extremophilic Preference?

**DOI:** 10.3390/life11080773

**Published:** 2021-07-30

**Authors:** Ádám Radványi, Ádám Kun

**Affiliations:** 1Department of Plant Systematics, Ecology and Theoretical Biology, Institute of Biology, Eötvös Loránd University, 1053 Budapest, Hungary; 2Institute of Evolution, Centre for Ecological Research, 1121 Budapest, Hungary; adam.kun@ttk.elte.hu; 3Parmenides Centre for the Conceptual Foundation of Science, Parmenides Foundation, 82049 Pullach, Germany; 4MTA-ELTE Theoretical Biology and Evolutionary Ecology Research Group, 1117 Budapest, Hungary

**Keywords:** mutational robustness, standard genetic code, genetic code origin, evolution of the genetic code, origin of life, extremophiles, codon usage, GC-content, environmental selection

## Abstract

The genetic code was evolved, to some extent, to minimize the effects of mutations. The effects of mutations depend on the amino acid repertoire, the structure of the genetic code and frequencies of amino acids in proteomes. The amino acid compositions of proteins and corresponding codon usages are still under selection, which allows us to ask what kind of environment the standard genetic code is adapted to. Using simple computational models and comprehensive datasets comprising genomic and environmental data from all three domains of Life, we estimate the expected severity of non-synonymous genomic mutations in proteins, measured by the change in amino acid physicochemical properties. We show that the fidelity in these physicochemical properties is expected to deteriorate with extremophilic codon usages, especially in thermophiles. These findings suggest that the genetic code performs better under non-extremophilic conditions, which not only explains the low substitution rates encountered in halophiles and thermophiles but the revealed relationship between the genetic code and habitat allows us to ponder on earlier phases in the history of Life.

## 1. Introduction

The origin of the genetic code and translation remains amongst the greatest conundrums of Life despite more than 50 years of research [1]. There are many actively debated, sometimes opposing, theories trying to explain it [2,3,4,5,6,7,8,9]. One possible way to crack the code is to study how mutations affect the integrity of proteins and how this might depend on the environment or genomic properties.

One of the key features of the standard genetic code (i.e., the nearly universal mapping of amino acids to codons) is its capacity to minimize the deleterious effects of mutations, thus making it optimized for translation [8,9,10,11]. The basis for this argument is the average fitness cost of replacing one amino acid with another due to mutation, measuring the change on a scale of amino acid physicochemical properties (e.g., hydrophobicity or polarity) that should correspond to the risks of misfold and loss of function in proteins [8,9]. Several studies have pointed out that the overwhelming portion of random alternative genetic codes have inferior error minimization capacities (i.e., lower mutational robustness) and the effect of a mutation is expected to be more detrimental. In comparison, only a few variants show that the code can be locally improved with codon reassignments [12,13]. This leads to the claim of error minimization playing a significant (but not exclusive [14]) role in the evolution of the standard genetic code [8,9,15].

Unfortunately, this is only one piece of the puzzle. For the actual genetic code, the average effect of mutations depends on codon usage, as certain codons carry an increased or decreased risk of deleterious effects when mutated [16,17,18,19]. For example, compared to AGY codons, the mutation of UCN codons of Ser is less likely to lead to major shifts in physicochemical properties of the mutant protein.

Codons, generally, are not employed in equal frequencies. The variability in codon usage comes partly from amino acid usage that differs from the frequencies dictated by the standard genetic code. Differential usage of amino acids could stem from the environment, as the general requirements for functional proteins may change with the habitat. The most renowned examples of environmental selection are studied in halophiles and thermophiles possessing characteristic amino acid distributions and codon usage in order to retain functional protein structure [20,21,22,23,24,25,26,27,28,29,30,31,32,33] by preferring specific interaction types or molecular strategies [25,34,35,36,37,38,39,40,41,42,43,44,45]. Other specialists, such as alkaliphiles, acidophiles and barophiles, can also possess recognizable patterns [28,46,47].

Secondly, there is a strong correspondence between codon usage bias and GC-content [32,48,49,50,51]. The exact causes of its variation remain an ongoing area of research; for example, bias in nucleobase composition has been associated with specific lifestyles [52,53,54,55,56] and mutational bias [57,58,59,60], whereas it appears that high GC-content in protein coding regions is not necessarily an adaptation to high temperature [61]. In any case, we must account for the predominant GC-effect.

After determining these two factors, namely environmental selection and GC-content, we ask the following questions: Can their effects on codon usage have further repercussions with respect to mutational robustness (i.e., the average effect of mutations in proteins)? In what condition does the genetic code minimize the effect of mutations considering the observable biological variance of codon usages?

The change in mutational robustness along these codon usage determining gradients can be studied using the information theoretic metric of distortion [16,17,18,62,63,64]. The measure of distortion contains the same essentials as previous well-known measures of the field [8,9,10], that is, (i) the estimated probabilities of non-synonymous mutation events occurring within the genome (described in a background mutation model); and (ii) the estimated cost of such missense mutation in proteins (one amino acid translated as another), based on a physicochemical trait (e.g., hydrophobicity). However, contrary to previous measures, distortion also builds in a third term (iii) by weighting for codon usage. This results in the average change in said amino acid physicochemical property caused by a non-synonymous mutation, given a distribution of codon usage.

The purpose of this study is to analyse the response of the average effect of genomic mutations associated with different environments (temperature, NaCl concentration and pH) and genomic gradients (GC-content) that govern codon usage. Using two comprehensive datasets comprising codon usage, genomic and environmental data from all three domains of Life, we estimate the expected severity of non-synonymous mutations by calculating distortions in four physicochemical properties: hydropathy, polar requirement (both related to hydrophobicity), molecular volume (i.e., the size of the amino acid), and isoelectric point (related to charge and polarity). We focus on their change with said environmental and genomic covariates.

Our findings suggest that the genetic code shows a certain preference towards non-extremophiles, that is, the codon usage of these organisms is expected to be more robust against mutations. In comparison, fidelity in the studied physicochemical properties can decrease in extremophiles, especially in thermophiles. This not only helps in explaining the low substitution rates found in extremophilic taxa, but also allows us to reflect on the relationship between habitat and the genetic code, and how it might help us to reconstruct earlier phases in the history of Life.

## 2. Materials and Methods

### 2.1. Data Aquisition

The nucleobase and codon distributions of each organism’s coding region were established from the Uniprot Reference Proteome database, which provides a representative cross-section of the taxonomic diversity found within UniProtKB [65]. Using NCBI Taxonomy ID-s [66], this codon usage dataset was cross-referenced with databases containing optimal environmental conditions. In one dataset, the environmental data for optimal growth temperature, pH, and salt concentration were obtained from the BacDive database [67] for 402 prokaryotes (Appendix A). A second environmental database was made available by Engquist [68], which resulted in a compilation of optimal temperatures for 3873 taxa, including Eukaryotes (Appendix A).

### 2.2. Distortion as a Measure for Code Performance

Distortion (Equation (1)) is used to estimate the average effect of mutations [16,17,18,69]. It is calculated given a source distribution of codons, *P(c_i_)*, and the uncertainty of the code resulting from noise, *P (Y = c_j_|X = c_i_)*, i.e., the probability of codon *c_i_* mutating into *c_j_* (see next section about background mutation model). Another important element of distortion is the distortion matrix with elements *d(aa_i_,aa_j_)* of physicochemical properties. Distortion matrices are essentially identical to “error matrices” widely used by other studies of random genetic codes (e.g., [8,9]). A distortion matrix *d* with elements *d(aa_i_,aa_j_)* specifies the cost associated with mistaking the encoded symbol *aa_i_* (amino acid of codon c_i_) of the source (genome *X*) and reproducing it as *aa_j_* in the replicated copy (genome *Y*). In our case, this cost is the absolute change in a specified physicochemical property. We define *d(aa_i_,aa_j_)* = 0 if *aa_i_ = aa_j_*, that is, *c_i_* and *c_j_* codes for the same amino acid. Mutations to stop codons are forbidden.

In short, the term Σ_i,j_ *P (Y = c_j_|X = c_i_) × d(aa_i_,aa_j_)* is essentially the same as earlier cost functions used to define the “error-minimization” or “mutational robustness” of genetic codes (for reference, see Freeland and Hurst [8] or Haig and Hurst [9]). Distortion adds a new term by weighting for codon distribution *P(c_i_)*.
(1)$$D=\underset{i,j}{\sum }P(c_{i})\times P\left(Y=c_{j}|X=c_{i}\right)\times d\left(aa_{i},aa_{j}\right)$$

Multiple distortion matrices were defined in order to measure different physicochemical dimensions of code performance. For these, we use a set of properties made available by Haig and Hurst [9], which include hydropathy, polar requirement, molecular volume and isoelectric point, yielding four different measures of code performance, denoted as *D*_Hyd_*, D*_Pol_*, D*_Vol_ and *D*_pI_. Note that PAM [70], BLOSUM [71] and related empirical matrices are not appropriate as they incorporate additional information with regard to the structure of the genetic code [72].

### 2.3. Background Mutation Model

The conditional probabilities *P (Y = c_i_|X = c_j_)* are the result of random mutations appearing in the genome and describe the chance of triplet *c_i_* mutating into triplet *c_j_* (insertions and deletions are ignored for now). In order to approximate these probabilities, a background mutation model is required to describe the generalized mechanism for spontaneous DNA mutations. We must estimate the raw, *a priori* performance of the genetic code without natural selection introducing additional bias. To this effect, we employ a simple model of random mutations [69] (Equations (2)–(4)), which is reminiscent of Kimura’s two parameter model [73]. Here, *κ* denotes the transition/transversion rate ratio, otherwise known as the ti/tv-ratio, and *µ* is the mutation rate. The inherent structure of the genetic code defines the probability of which codon *i* mutates into codon *j* given the occurrence of a transition or a transversion; these are denoted by terms *P (c_i_ → c_j_|*ti*)* and *P (c_i_ → c_j_|*tv*)*, respectively. Expected proteomic distortions were then calculated for each taxon’s codon composition. Since our goal is a comparative analysis between taxa, the effect of *µ* is unimportant.
(2)$$\kappa =\frac{p_{\mathrm{ti}}}{p_{\mathrm{tv}}}$$
(3)$$P\left(Y=c_{j}|X=c_{i}\right)=µ\times \left[\frac{\kappa }{\left(1+\kappa \right)}\times P\left(c_{i}\to c_{j}|\mathrm{ti}\right)+\frac{1}{\left(1+\kappa \right)}\times P\left(c_{i}\to c_{j}|\mathrm{tv}\right)\right]$$
(4)$$P\left(Y=c_{i}|X=c_{i}\right)=1-µ$$

First, a detailed study of the model is carried out at a specific transition–transversion ratio (*κ* = 2.5), as this rate is close to those observed in a number of studies [57,58,74]. However, this ratio encountered in non-coding regions could still be the result of a remaining, albeit relaxed, error correction or negative selection. It can also vary greatly from organism to organism [75], with rates close to uniformity (*κ* ≈ 0.5) [76], or very strong bias towards transitions (*κ* ≈ 10) [77]. To ensure the robustness of our analysis, distortions are also calculated for an interval of ti/tv-ratios. The resulting measures are, therefore, *D*_Hyd_(*κ*)*, D*_Pol_(*κ*)*, D*_Vol_(*κ*) and *D*_pI_(*κ*), where *κ* = [0.5, 10] (Appendix A).

### 2.4. Multi-Linear Regressions

On both datasets, we performed multi-linear regression in RStudio [78]. Response variables *D*_Hyd_(*κ*), *D*_Pol_(*κ*), *D*_Vol_(*κ*) and *D*_pI_(*κ*) were modelled separately using the available environmental variables as covariates. The genomic content of guanine and cytosine (GC-content) was also included as a quadratic polynomial predictor (based on other works using information theoretic metrics [51]) to control for its predominant effect on the codon composition [48]. The regressions were applied for each ti/tv value individually and the influence of ti/tv on the robustness of predictor effects was then determined by plotting the partial effects (*β*) of each covariate against *κ*.

## 3. Results

We analyse the mutational robustness of codon usage profiles associated with different codon frequency determining factors. To this effect, we prepared two different datasets: Appendix A comprises codon usage data for 402 prokaryotes, and includes environmental data on optimal growth temperature, pH, and salt concentration, whereas Appendix A contains a larger sample of 3873 archaeal, bacterial and eukaryotic taxa, but only temperature as an optimal environmental condition. Then, we calculated distortions (i.e., the average effect of a non-synonymous mutation) in four distinct physicochemical properties (hydropathy, polar requirement, molecular volume, and isoelectric point). We employ multi-linear regression models to study the response of distortions to environmental covariates and GC-content.

Here, we report that both environmental selection and GC-content can affect codon usage in ways that are expected to impact mutational robustness. This leads to increased distortions, that is, reduced physicochemical fidelities, in thermophiles and halophiles.

In all cases of the four distinct physicochemical distortions, the overall regressions are significant (*p* < 0.001). Both datasets show relatively high *R^2^* values for each attribute, especially in the cases of distortions of hydropathic and isoelectric properties (Table 1 and Table 2, also including the relative effects as standardized partial coefficients (β_std_) where variables were Z-transformed prior to analysis). These general results indicate that our regression models provide good fits. In the following, we discuss the effects of environmental selection and GC-bias on distortions separately.

### 3.1. Environmental Factors Mainly Increase Expected Distortions

The regressions imply a detailed relationship between the expected distortions and environmental selection (Table 1, Figure 1), especially temperature (Table 2, Figure 2). Temperature has significant distortive effects, first demonstrated by the prokaryotic dataset (Appendix A) showing a significant increase in distortions with optimal growth temperature (Table 1; hydropathic distortion: *β*_std_ = 0.16; *t* = 7.97; *p* < 0.001; distortion in polar requirement: *β*_std_ = 0.12; *t* = 3.02; *p* = 0.003; volumetric distortion: *β*_std_ = 0.42; *t* = 9.06; *p* < 0.001; distortion in isoelectric pattern: *β*_std_ = 0.32; *t* = 10.07; *p* < 0.001). These effects are also comparable with the distortive effects of temperature observed in the larger sample of Appendix A (Table 2; hydropathic distortion: *β*_std_ = 0.15; *t* = 21.51; *p* < 0.001; distortion in polar requirement: *β*_std_ = 0.16; *t* = 11.92; *p* < 0.001; volumetric distortion: *β*_std_ = 0.33; *t* = 22.31; *p* < 0.001; distortion in isoelectric pattern: *β*_std_ = 0.17; *t* = 14.37; *p* < 0.001); thus, Eukaryotes fit in well with the trends observed in prokaryotes (Figure 2). To summarize, distortions increase in codon compositions adapted to high temperature, so genomic mutations are prone to produce more severe amino acid substitutions with respect to all four examined physicochemical properties.

The prokaryotic data of Appendix A also allows us to include the effects of halophilic adaptations and ambient pH in regressions (Table 1, Figure 1). Organisms adapted to higher NaCl concentrations possess codon compositions less prone to hydropathic distortion (*β*_std_ = −0.10; *t* = –5.25; *p* < 0.001), but the average effect of incidental mutations is predicted to be more severe if distortion is measured via polar requirement (*β*_std_ = 0.36; *t* = 9.43; *p* < 0.001). These opposite effects are somewhat surprising, as both hydropathy and polar requirement measure the hydrophobic character of amino acids and have high correlation (*r* = –0.79; *t* = –5.39; *p* < 0.001).

The reason for the ambiguous effect of halotolerance on hydrophobic characters is that Glu and Asp are highly favoured in halotolerant proteins [28,30,79]. These acidic amino acids show large discrepancies in hydropathy and polar requirement [9]; therefore, a comparison of related distortions will be also sensitive to the usage of Glu and Asp codons. The relative position of halophilic archaea supports this reasoning (top two plots in the middle of Figure 1).

Nevertheless, hypersaline environments select for codon usages that have a reduced fidelity in polar and charge patterns, probably the most important features in halophiles (see Discussion). This is shown by the increased distortion of isoelectric properties (*β*_std_ = 0.21; *t* = 6.86; *p* < 0.001).

The effects of ambient pH remain inconclusive; regardless of *p*-values, the standardized beta coefficients indicate that effect sizes are negligible (hydropathic distortion: *β*_std_ = 0.06; *t* = 2.94; *p* = 0.003) compared to other environmental factors, or non-significant. This can be attributed to the relative invariance of intracellular pH regardless of the ambient environment [80].

### 3.2. Robustness of Enviromental Effects with Regard to Ti/Tv-Ratio

Choosing the parameter of *κ* = 2.5 in our background mutation model to calculate distortions is a biologically reasonable assumption [57,58,74]. However, to see how this assumption could have influenced our study, we extend the analysis to a wider range of ti/tv-ratios using the data comprising prokaryotes and their respective optimal growth environment (Appendix A). We recalculate distortions in an interval of ti/tv-ratios (*κ* = {0.5, …, 10}), reemploy the multi-linear regression models, and plot the partial effect (*β*) of each covariate against *κ* to see how our assumption of *κ* could influence the interpretation of the environmental effects on distortions.

The results are shown on Figure 3. The partial regression coefficients at increasing κ are qualitatively similar, the effects of environmental factors change but only asymptotically, and the signs of these coefficients do not change in general. The only exception is the effect of temperature on the expected distortion in polar requirements. Here, in higher ranges of ti/tv (*κ* > 3.5), the distortive effect becomes non-significant (*p* > 0.01). Nonetheless, we may conclude that the other effects of environmental selection remain similar over a broader range of ti/tv-ratios; thus, our interpretations of the impact of environmental conditions remain valid, including the distortive effect in thermophiles with regard to the other three physicochemical traits.

### 3.3. The Effect of GC-Content on Physicochemical Fidelities

The inclusion of quadratic effects on GC-content is mainly used to control for more elusive effects generating shift in genomic composition (see Introduction). All distortions on Figure 1 and Figure 2 show significant variations in the mesophilic range. This is attributed to the predominant effect of GC-bias (Table 1 and Table 2).

In the larger sample of Appendix A, which includes only optimal temperature data, all distortion measures show a significant quadratic response to GC (Table 2). Said significant quadratic responses in Table 2 are caused presumably by an unaccounted effect of salt concentration leading to high GC-bias typical of halophilic archaea [30] (observe the distortion in polar requirements on Figure 2, and the cluster of archaea around 37.5 °C) or other unaccounted effects.

This is further supported by the fact that most effects of GC-content are missing if we account for all three environmental factors in Appendix A (Table 1). Only the distortion in polar requirements stays quadratic in this case. The effect on hydropathic distortion appears linear, with the quadratic term remaining non-significant. We also find a weaker statistical support for a quadratic response of distortion in isoelectric properties (*β*_std_ = –0.73; *t* = –2.27; *p* = 0.024), indicating higher fidelity in GC-biased proteomes. The response of volumetric distortion is not significant. Just as in the case of environmental effects (Section 2.3), these effects (or the lack of thereof) are largely robust to ti/tv-ratio (Figure 4).

To visualise the minima or maxima arising in the assumed relationship between GC-content and distortions at different ti/tv-s (*κ* = {0.5, …, 10}), we have extrapolated from the available data to the whole parameter space. Data were extracted, similar to the process described in Section 3.2, using the prokaryotic data controlling for all environmental variables (Appendix A). Then, expected distortion values were standardized for each value of *κ*. Figure 5 shows the results, indicating that the expected (fitted) distortion in the hydropathic trait is lowest at low GC-contents. Conversely, the maximum fidelity (i.e., minimized distortion) measured in polar requirements is achieved with a moderately high GC-bias, along with a slight shift of minima towards intermediate GC-content at increased *κ*. These effects of GC-content on hydrophobicity-related distortions (hydropathy and polar requirement) appear at first sight to be antagonistic, cancelling each other out. However, as the regression of the hydropathic distortion results in better fit compared to polar requirement (Table 1), we suggest that the general fidelity in hydrophobic property is maximized at low-intermediate or intermediate GC-content. Nonetheless, this relationship will require further analysis in the future.

## 4. Discussion

To figure out the evolution of the genetic code, we must also understand how factors known for affecting codon or amino acid usage, such as environmental selection [20,21,22,23,24,25,26,27,28,29,30,31,32,33] and the GC-composition of the coding region [48,51,81,82], may leave their mark on code usage. We can estimate the average impact of non-synonymous genomic mutations (i.e., distortion) via the change in four distinct amino acid physicochemical properties: hydropathy, polar requirement (both are hydrophobic characters), volume, and isoelectric point (the latter reflecting polar properties and charge). With more whole genomes available and datasets on optimal growth conditions (optimal growth temperature, NaCl concentration and pH), we can establish the expected change in distortions with optimal environmental conditions and GC-content.

The current study confirms that extreme environments, by selecting for specific codon usages, can negatively affect physicochemical fidelity, increasing the average severity of incidental genomic mutations. This is demonstrated on two datasets: (i) one dataset documents optimal temperature, salt concentration, and pH conditions for Bacteria and Archaea [67], whereas the (ii) second compiles optimal growth temperature conditions for a more comprehensive sample including members of all three domains [68]. Although Bacteria are overrepresented in both, our earlier analysis on a smaller but phylogenetically controlled dataset showed similar effects that are robust to taxon sampling [69]. We can also confirm that the inclusion of Eukaryotes is unlikely to change this picture. Thus, at least with regard to habitat temperature, we may be revealing a trend that is universal to all organisms of Earth: environmental selection is likely to affect the efficiency of using the standard genetic code.

### 4.1. Selection in Extremophiles can Decrease Mutational Robustness

Adaption to increasing temperatures results in codon usage profiles that are generally worse at maintaining hydrophobic, polar, and volumetric patterns, than those of organisms (including Eukaryotes) thriving at lower temperatures. In other words, the conservation of these physicochemical properties tends to be more unreliable and the effects of incidental mutations are expected to be more severe. Despite the adjustment of codon usage against such mutations [28], this reduced mutational robustness of thermophiles could lead to an increased risk of defective mutants (e.g., by disturbing folding processes or decreased stability), especially if we consider the higher importance of hydrophobic interactions and salt bridges in thermoadapted proteins compared to mesophile orthologs [29,37,42,83].

The ultimate effect of selection for halotolerance is similar. Although the exact impact on hydrophobic characters is hard to assess, the codon usage profile of halophiles has clear costs of higher distortion of polar and charged (isoelectric) properties. Halophilic proteins are often characterized by ion-pair networks, and a higher abundance of acidic residues is required to grant solubility at high intracellular K^+^-concentrations while hydrophobic cores become weaker [25,34,38,39]. These strategies suggest that the importance of polar properties outweigh other effects (including hydrophobic fidelity). Hence, codon usage of halophiles might be just as suboptimal as thermophiles at retaining biologically relevant physicochemical patterns.

These observations may partly explain why thermophiles [84,85,86] and halophiles [87,88] possess remarkably low mutation rates compared to other taxa. The strong selection patterns against mutations primarily encountered in these extremophiles might be caused by the relative inefficiency of codon usages in these environments, for which the genetic code seems especially ill-suited with regard to mutational robustness. Therefore, in these species, the increased fidelity of replication could exist as an adaptation to avoid these harsh fitness costs posed by the more adverse effects of genomic mutations. To put it differently, if mutations have milder fitness consequences, then higher mutation rates are tolerated (which is the case for mesophiles). Per generational mutation rates, for example, are higher for longer living organisms having lower population sizes (such as ourselves), compared to short lived, but numerous organisms (such as bacteria) because lower population sizes make some of the slightly deleterious mutations nearly neutral [89,90].

### 4.2. GC-Content and Its Effect on Distortions

For the effect of GC-content, our models predict significant effects in distortions of hydrophobic attributes (hydropathy and polar requirement). The hydrophobic core and hydrophobic interactions are generally regarded as the primary guides of protein folding [91,92,93], further supplemented by the fact that secondary structures can often be described and predicted according to these properties [94]. Therefore, our results suggest that GC-bias (and related factors) might have a secondary impact on mutational robustness.

Unfortunately, the exact effect on hydrophobicity is difficult to entangle in this study, albeit a slightly pronounced AT-bias is the most likely to minimize distortion. This could be in accord with an earlier analysis accounting for phylogenetic bias in a small sample of prokaryotes [69], and other observations attesting for a general AT-biased substitution pattern [57,58,59]. If that is the case, an A/T-mutation should be more likely to fix due to its lower risk of perturbing hydrophobic patterns in proteins. For the record, however, we mention that there are other mechanisms consistent with AT-bias [95], and the antagonistic responses of hydropathy and polar requirement found in this study demand further analysis.

Other effects of genomic GC-composition could indicate more specific needs. We also find that a higher usage of G and C can minimize distortion in isoelectric properties. As discussed earlier, the molecular strategy of halophiles might require lower distortion in this feature, while the underlying environmental selection acts against that. In this regard, the elevated GC-bias typical of halophiles [25,30] is likely to be a resulting side-effect of an already strong counterbalancing pressure to mitigate the distortive effects of mutations by exploiting the degeneracy of the genetic code along with the flexibility in the choice of physiochemically similar amino acids. The remaining negative effects along the salinity gradient (Figure 1) could show the insurmountable limits of the mapping. This again emphasises that causal relationships between the nucleotide and amino acid compositions cannot be completely separated; they depend on various evolutionary and environmental factors [26,32,45,82,96,97,98,99] and there are additional trade-offs between these two, especially in genomes with more extreme nucleotide compositions [51].

### 4.3. Limitations of the Current Model and Future Prospects

As with all computational models, our framework also works with simplifying assumptions. Here, we focused on the effects propagated by genomic mutations. On this level, the probability that a triplet *c_i_* is substituted by *c_j_* does not depend on which base is mutated, whereas misread probabilities clearly depend on the position in the event of mistranslation [8,18,100], which is an additional, albeit not constant and not hereditary, level of distortion.

Another obvious caveat is the assumption that translation mechanisms are uniform across the Tree of Life. Although the genetic code is quasi-universal, the entire apparatus of protein synthesis is not. There could be many species-specific adaptations to compensate for environmental conditions or other codon usage defining effects, for which our analysis cannot account. In this regard, we have already proposed the lowered mutation rate as a likely environment-specific mechanism to optimize translation in certain extremophiles, but other species-specific adaptations could exist to compensate for environmental conditions or biased genomic composition.

We also operate under the assumption that all proteins are expressed at the same level. That is an obvious simplification. The expression levels have evolutional importance [101] related to thermophilic properties [102] and misfold chance [103], emphasizing that the genetic code cannot be understood without the other structural aspects of proteins (e.g., change in folding free energy [18,100]). We also note that, compared to other prokaryotes, thermophiles and halophiles possess elevated rates of horizontal gene transfer [104,105,106]. Using complete proteomes might interfere with the analysis due to recently acquired genes in these taxa, and a succeeding study focusing on core genes is needed. We believe that upcoming models will be able to address some of the issues by comparing the mutational effects on more local structures (e.g., active sites or aligned sequences). In this regard, employing mixed models on a small subset of protein families responsible for core functions (e.g., translation, transcription, and replication) is already underway.

### 4.4. Implications of the Non-Extremophile Optimality of the Genetic Code and Codon Usage

The evolution of genetic code was a long process likely to be affected by multiple driving forces (e.g., stereochemical affinity between codons and attributed amino acids [4,5] or coevolution between the biosynthetic paths of amino acids and cognate codons [6,7]). From start to finish, it cannot be understood solely by natural selection increasing mutational robustness [14]. At first, this feature might have been only a consequence of evolution based on other mechanisms [107,108,109,110,111]. Notwithstanding, the definitive version of the genetic code appears to be a near-optimal robust mapping, because the majority of, but not all [12,13], alternative codes fall short of such error minimalizing capacity [8,9]. Hence, the environment preference of the genetic code could potentially help us to reconstruct its late phase of development along with the conditions that might have witnessed its finalization.

Our most striking result is that this final snapshot of the genetic code seems to be less compatible with codon usages associated with high temperature. In terms of accuracy, its organization prefers a moderate, mesophilic environment instead. This observation not only contradicts earlier analyses suggesting extremophilic preference of the code [46,47,112,113], but also seem to go against influential phylogenies that provided intuitive evidence of an extremophile LUCA by placing thermophiles as the most basal groups [114,115,116,117].

The common thermophilic ancestor, however, is only one side of the coin. It facilitates a somewhat overreaching logic that the cradle of life, including the development of the genetic code, was always associated with “infernal” environments of the Archean Earth. Even the alleged thermophilic nature of LUCA is questionable, as rRNA and protein sequences indicate that hyperthermophilic features of basal clades are actually parallel adaptations, while the original root could have been non-thermophilic [118,119,120,121].

Whether LUCA was thermophilic or not, the standard genetic code is likely to give a more complete picture of the story. LUCA was not the first living organism, nor the one in which the genetic code evolved. Considerable time and environmental change may have occurred between the first fully peptide-DNA based, translationally capable cell and LUCA. The Late Heavy Bombardment (~4 Gyr ago), around which the LUCA was recently dated [122], could be one probable cause of thermophilic lifestyles appearing as an ancestral feature. Such a global cataclysm might have eradicated all surface life adapted to moderate temperatures, leading to an early extinction event and a lack of information on an ancient mesophilic biosphere favoured by the genetic code. There is, moreover, tentative evidence of a period with a cooler Earth having a solid crust and liquid water 4.5 Gyr ago [123] due to the relatively lower output of the young Sun [124,125], which could support the existence of such early moderate environments. Our results are entirely consistent with this scenario, opening up the possibility of a mesophilic phase in the early history of Life.

We must, however, emphasize that our results do not in any way imply that Life or the primordial genetic code originated in a cold or mesophilic environment. We analysed a final state of the genetic code that could only provide a glimpse into an era of its final stage, much later than the Origin of Life or a primordial genetic code, but somewhere before LUCA. We assume only that the canonical genetic code was “frozen” while it documented the ruling conditions. This period could have been preceded and followed by periods with significantly different environmental conditions.

Additionally, there is another aspect of mesophilic optimality that should hold even if our naive conjectures are false. This pertains to the history after LUCA and involves the perks of relaxed selection in mesophilic environments due to its better fit with the genetic code. The distortions of mesophilic codon usages have shown that they are not as sensitive to mutations. This not only means that low temperature ranges allow for a much more economic replication, e.g., less sophisticated replicases or proofreading capacities leading to higher mutation rates. In these environments, it also becomes easier to explore the sequence space of proteins without serious fitness consequences, whereas without mesophilic niches, the structural constraints [126] and low substitution rates required at higher temperature [84,86] should have hindered the long-term evolvability and the colonization of variable environments [127].

Somewhere after LUCA, the environment cooled gradually in the interval from 3.5 to 0.5 Gyr ago according to an analysis of elongation factors [128], allowing the offspring of LUCA to adapt and colonize colder niches. This might have happened at a relatively fast pace thanks to this specific code that prefers a mesophilic environment. We can only imagine how differently (e.g., at what other rate) Life might have evolved and radiated with a different codon to amino acid mapping.

## Figures and Tables

**Figure 1 life-11-00773-f001:**
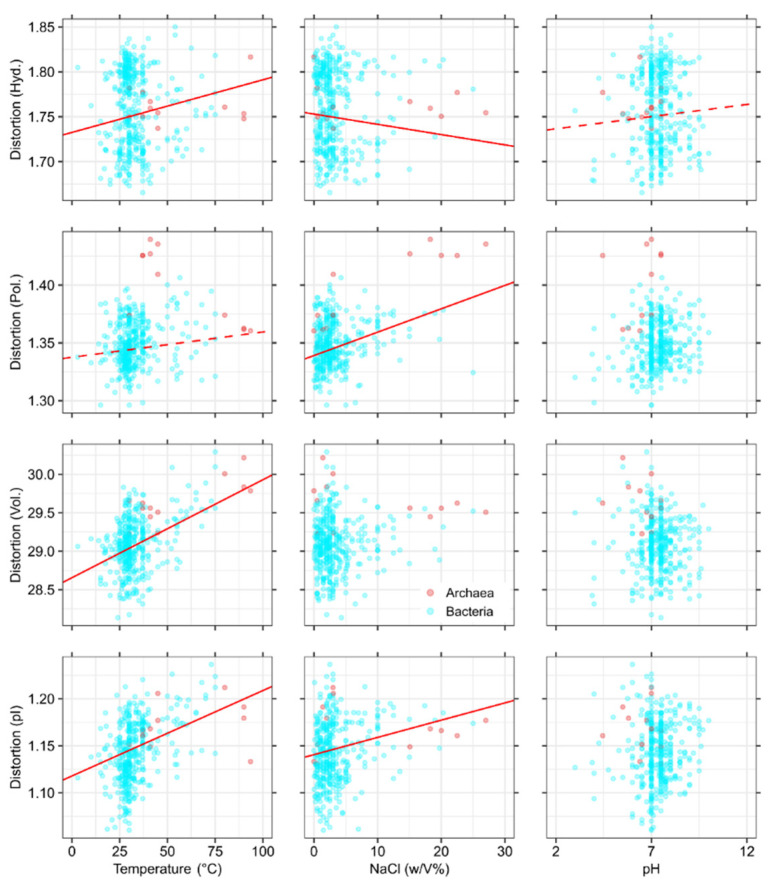
The effect of environmental variables on the expected distortion of amino acid physicochemical properties (hydropathy, polar requirement, molecular volume, and isoelectric point, respectively; *κ* = 2.5), using prokaryotic data (*n* = 402; Appendix A). The effect of optimal temperature on codon usage has clear distortive effects with respect to all four physicochemical traits. Hypersaline environments select for codon usages that have a reduced fidelity of polar and charge patterns, shown by the increased distortion related to isoelectric points (pI). Partial regression lines are drawn by setting remaining covariates as GC = 0.5; T = 30 °C; cc_NaCl_ = 2.5 *w*/*V*%; pH = 7 (continuous line: *p* < 0.001; dashed line: *p* < 0.01).

**Figure 2 life-11-00773-f002:**
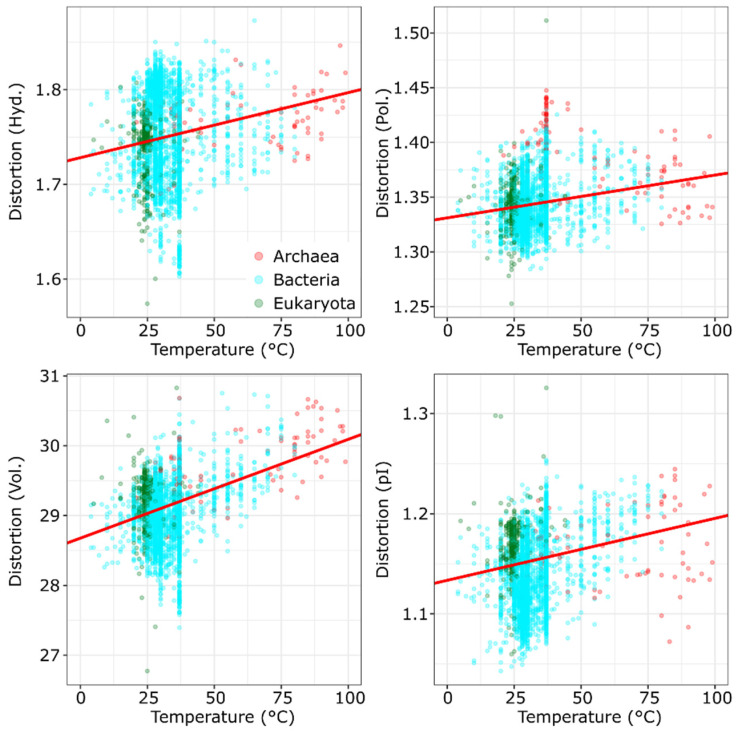
The effects of optimal growth temperature on the expected distortion of amino acid physicochemical properties (hydropathy, polar requirement, molecular volume, and isoelectric point, respectively; *κ* = 2.5), using proteomes comprising all three domains of Life (*n* = 3873; Appendix A). The effect of optimal temperature on codon usage has clear distortive effects with respect to all four physicochemical traits. Partial regression lines of temperature show responses adjusted for GC = 0.5 (*p* < 0.001).

**Figure 3 life-11-00773-f003:**
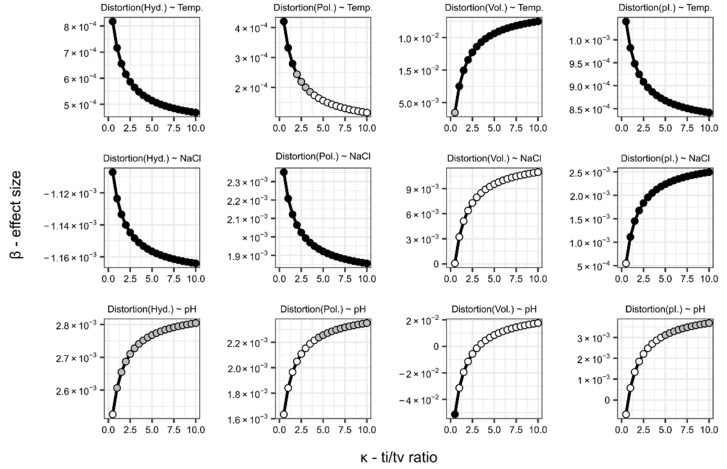
The effect of ti/tv-ratio (*κ*) on the effects of environmental coefficients (*β*) to the responses of the four distinct physicochemical distortion metrics. Different ti/tv-s are used in the background mutation model to re-calculate distortions and we measure the change in partial coefficients after re-employing regression. Distortion values are from the dataset comprising prokaryotes and their respective optimal growth environment (Appendix A). For reference, see Table 2 for the exact values at κ = 2.5. The significance level of partial regression coefficients is marked by shade (black: *p* < 0.001; grey: *p* < 0.01; white: *p* > 0.01).

**Figure 4 life-11-00773-f004:**
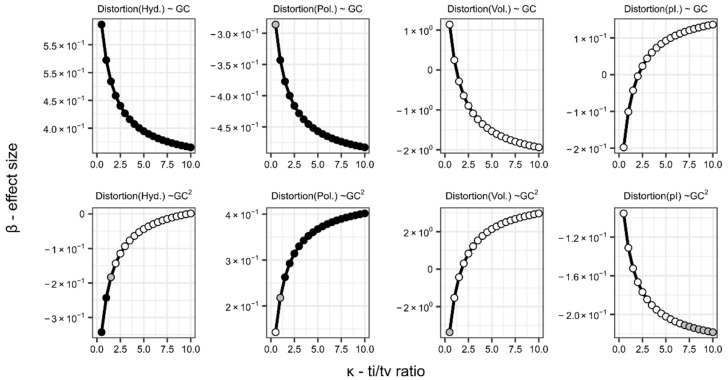
The effect of ti/tv-ratio (κ) on the coefficients (β) approximating the quadratic effect of GC-content to the responses of the four distinct physicochemical distortion metrics. Different ti/tv-s are used in the background mutation model to re-calculate distortions and we measure the change in partial coefficients after re-employing regression. Distortion values are from the dataset comprising prokaryotes and their respective optimal growth environment (Appendix A). For reference, see Table 2 for exact values at κ = 2.5. The significance level of partial regression coefficients is marked by shade (black: *p* < 0.001; grey: *p* < 0.01; white: *p* > 0.01).

**Figure 5 life-11-00773-f005:**
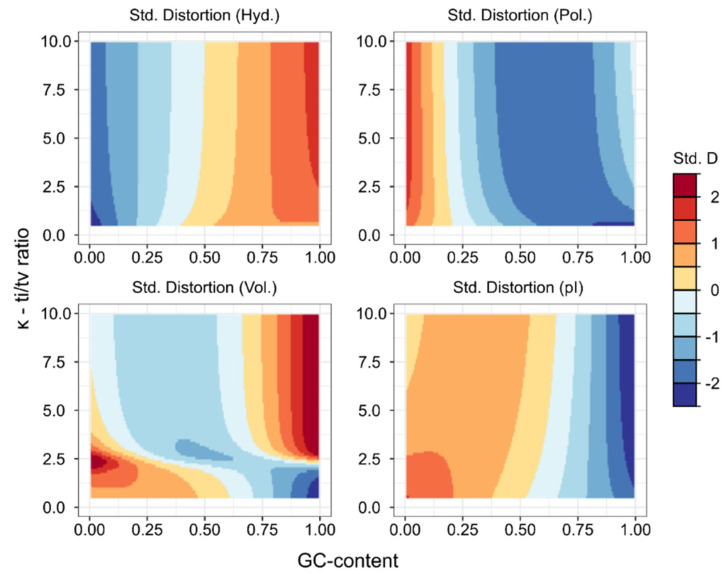
The extrapolated responses of physicochemical distortions (hydropathy, polar requirement, molecular volume, and isoelectric point, respectively) to the effects of GC-content at different ti/tv-ratios. All distortions are controlled for environmental variables from the prokaryotic dataset (Appendix A). The values of predicted distortions were standardized at every *κ* = {0.5, …, 10}.

**Table 1 life-11-00773-t001:** Results of the four multi-linear regressions performed on prokaryotic distortions (*n* = 402; Appendix A) and the effects of environment (optimal growth temperature, NaCl concentration, pH) and GC-content on these physicochemical distortions (*D*). Distortions (in hydropathy, polar requirement, molecular volume, and isoelectric point, respectively) and GC-contents are calculated from codon usage data of the Uniprot Reference Proteome Dataset [65]. Optimal growth environment data of prokaryotes are from the BacDive database [67].

D	Parameter	Unstandardized Coeff.		*Β* _std_	*t*	*p*	*F* _(5,397)_	*R^2^*	*p*
		β	SE						
Hyd.	Intercept	1.525	0.018	-	82.653	0.000	471.286	0.854	0.000
	Temp.	0.001	0.000	0.160	7.967	0.000			
	NaCl	−0.001	0.000	−0.100	−5.247	0.000			
	pH	0.003	0.001	0.060	2.944	0.003			
	GC	0.440	0.067	1.300	6.529	0.000			
	GC^2^	−0.115	0.063	−0.360	−1.810	0.071			
Pol.	Intercept	1.447	0.018	-	79.729	0.000	61.372	0.430	0.000
	Temp.	0.000	0.000	0.120	3.024	0.003			
	NaCl	0.002	0.000	0.360	9.426	0.000			
	pH	0.002	0.001	0.090	2.327	0.020			
	GC	−0.416	0.066	−2.470	−6.275	0.000			
	GC^2^	0.314	0.062	1.980	5.026	0.000			
Vol.	Intercept	28.919	0.353	-	82.024	0.000	18.882	0.193	0.000
	Temp.	0.013	0.001	0.420	9.060	0.000			
	NaCl	0.007	0.004	0.080	1.744	0.082			
	pH	−0.006	0.018	−0.010	−0.327	0.744			
	GC	−0.898	1.288	−0.330	−0.697	0.486			
	GC^2^	0.830	1.213	0.320	0.684	0.494			
pI	Intercept	1.130	0.023	-	49.960	0.000	130.303	0.617	0.000
	Temp.	0.001	0.000	0.320	10.067	0.000			
	NaCl	0.002	0.000	0.210	6.857	0.000			
	pH	0.002	0.001	0.060	1.954	0.051			
	GC	0.023	0.083	0.090	0.282	0.778			
	GC^2^	−0.177	0.078	−0.730	−2.273	0.024			

**Table 2 life-11-00773-t002:** Results of the four multi-linear regressions performed on proteomic distortions comprising all three domains of Life (*n* = 3873; Appendix A), and the effects (*β*) of optimal growth temperature and GC-content on these physicochemical distortions (*D*). Distortions (in hydropathy, polar requirement, molecular volume, and isoelectric point, respectively) and GC-contents are calculated from codon usage data of the Uniprot Reference Proteome Dataset [65]. Temperature data are cross-referenced from [68].

D	Parameter	Unstandardized Coeff.		*Β* _std_	*t*	*p*	*F* _(3,3870)_	*R^2^*	*p*
		β	SE						
Hyd.	Intercept	1.531	0.005	-	284.970	0.000	5706.573	0.815	0.000
	Temp.	0.001	0.000	0.150	21.510	0.000			
	GC	0.468	0.021	1.370	22.490	0.000			
	GC^2^	−0.148	0.020	−0.450	−7.480	0.000			
Pol.	Intercept	1.436	0.006	-	259.990	0.000	557.517	0.301	0.000
	Temp.	0.000	0.000	0.160	11.920	0.000			
	GC	−0.324	0.021	−1.790	−15.140	0.000			
	GC^2^	0.228	0.020	1.320	11.210	0.000			
Vol.	Intercept	26.972	0.107	-	252.380	0.000	241.166	0.157	0.000
	Temp.	0.014	0.001	0.330	22.310	0.000			
	GC	6.031	0.413	1.890	14.590	0.000			
	GC^2^	−5.249	0.394	−1.730	−13.320	0.000			
pI	Intercept	1.128	0.007	-	155.120	0.000	1098.377	0.460	0.000
	Temp.	0.001	0.000	0.170	14.370	0.000			
	GC	0.175	0.028	0.650	6.230	0.000			
	GC^2^	−0.328	0.027	−1.270	−12.220	0.000			

## Data Availability

The data presented in this study are available in the Appendix A of this article.

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
