# Peer review of "The Mutational Robustness of the Genetic Code and Codon Usage in Environmental Context: A Non-Extremophilic Preference?"

_life, 2021, doi:10.3390/life11080773_

Round 1

Reviewer 1 Report

The mutational robustness of the genetic code and codon usage 2 in environmental context: a non-extremophilic origin?

Radványi AND Kun

This manuscript describes a complex analysis of the correlations between

aa usage, codon usage, and GC content in different species with the impact

of different environmental parameters that affect protein structure. These

correlations are then mapped along an axis that describes the average

environmental conditions for each species. Based of this analysis the authors

hope to determine the impact of translation errors for each of the species

studied, given their genomic structure and ecologic characteristics.

I must say, however, that the manuscript is so confusingly written that I

might be wrong even about the general purpose of the paper. In my

opinion, this manuscript needs to be thoroughly edited to be considered

for publication. The main problem is the confusing use of terms. For 

example:

line 32. 'Error capacity' for 'error tolerance'.

line 42 'by the genetic code' for 'by the translation machinery'

line 94 'reproduced copy (genome y)' for 'replication' (?) ', 'translation' (?)

lines 139-140. Meaning not clear.

And so on.......

I am not able to review the accuracy of the mathematical approach used

by the authors, so hopefully other editors will comment on its

appropriateness. However, in several instances the results are clearly not

reflecting the actual experimental data, which once again results in very 

contradictory statements such as those found in lines 282-289, or line 300

(how can low mutational rates be caused by a relative inefficiency of the 

genetic code ?).

In order to explain the contradictions between their calculations and

biological data the authors resort to incongruent arguments, such as:

- 'Fidelity of hydrophobic codons is controversial and obscure' (!)

- Thermophiles and halophiles have elevated LGT. (like so many other

prokaryotes......)

Although the authors try to describe the caveats that their work presents,

they ignore the most obvious one: they have assumed that translation

mechanisms are uniform across the tree of life. The genetic code is almost

universal, but the machinery for protein synthesis is not, and many species-

specific adaptations exist to compensate for environmental conditions or

biased proteome compositions. In other words, thermophiles have their own

mechanisms to optimize translation at high temperatures.

Finally, they conclude that their results preclude a thermophilic origin of life,

a statement that is just a rather naive projection of their observations. 

Genetic code fidelity need not be perfect at the first stages of development

of the translation machinery, so perhaps a more realistic conclusion of their

analysis would be that ' The impact of environmental parameters upon

translation fidelity the expansion of mesophilic species' ?. 

Reviewer 2 Report

This paper describes the analysis on the mutational robustness of the genetic code in the context of factors governing codon usage. By computational models and genomic and environmental data from all three domains of life, the authors estimated the expected severity of non-synonymous genomic mutations in proteins, and finally questioned against the influential phylogenies that have also provided an intuitive evidence of an extremophile LUCA by placing thermophiles as the most basal groups.

The authors’ viewpoint and approach are unique and I think it should be published finally. However, although I can understand their point, I do believe that general readers, especially with no mathematical background, can NOT follow the paper AT ALL in the present form. It is because of the total lack of the explanation of the background. The paper in the present form is not the reader-friendly. A lot of abbreviations appear in the text without explicit references to the original wordings. Especially, the explanation of Tables should be in more detail. In addition, schematic diagram of the logical flow of their whole research should be added at the beginning. Furthermore, the comparison of the mutational effects of some local structures of proteins (e.g., structures of specific active sites) should be performed as well as the whole genome sequences comparison. The last one is not mandatory because it may be a future prospect, but the authors should at least comment on the importance of the local structural comparison.

<Minor points>

Page 6 and 7, Figure 1 and 2

The colors corresponding to Archaea and Bacteria are hard to be discriminated, so one of the colors should be changed.

Page 20 and 21, Reference 119 and 123

Science (80-. ). -> Science

Page 13-21, Reference 19, 30, 41, 58, 69, 72, 78, 88, 89, 100, 102 and 113

Proc. Natl. Acad. Sci. -> Proc. Natl. Acad. Sci. U. S. A. (Only reference 7 is properly described.)

Reviewer 3 Report

The article of Radványi and Ku describes a study carried out on the origin of the genetic code and its evolution in different environments. The catchy title announces a non-extremophilic origin for the genetic code used in its current form. Using a battery of computational methods, the authors showed that the current genetic code is robust in mesophilic conditions but deteriorates in extremophilic conditions, especially in thermophiles. This result is ruling out the hypothesis of an extremophilic origin for the present-day genetic code.

To reach their conclusions, the authors have employed computational methods, considerable number of genomic sequences, databases on the habitats of microorganisms, and pre-existing equations to model mutation consequence and distortion. P-values have been used to validate the conclusions.

The progress of the study looks suitable however is very difficult to understand which parameters were tested. The presentation of results as tables and plots that are only partially interpreted does not help understanding. Generally, the manuscript would benefit of an extensive rewriting to engage a larger audience of readers. The evolution of the genetic code and the origin of translation is a fascinating problem that questions many biologists, chemists, or simply curious non-specialist readers. In the current form of the article, most readers will not be able to understand the experimental approach used, the distorted parameters and which databases were tested.

The article could greatly benefit from this revision and become accessible to a wider audience of readers.

Round 2

Reviewer 1 Report

I think that the manuscript is greatly improved, in terms of both readability and balance of the discussion. I remain skeptical about the significance of the results, but this is more a question of personal interpretation.

I wonder to what extent the effects detected by the authors are influenced by the three six-codon boxes of Leu, Arg, and Ser, and what would happen if these three aa were removed from the calculations or used alone.

There are still a few grammatical errors and typos that should be revised.
